# Association between IL-1 Gene Polymorphisms and Stage III Grade B Periodontitis in Polish Population

**DOI:** 10.3390/ijerph192214687

**Published:** 2022-11-09

**Authors:** Aniela Brodzikowska, Bartłomiej Górski, Agnieszka Bogusławska-Kapała

**Affiliations:** 1Department of Conservative Dentistry, Medical University of Warsaw, 02-097 Warsaw, Poland; 2Department of Periodontology and Oral Mucosa Diseases, Medical University of Warsaw, 02-097 Warsaw, Poland; 3Department of Comprehensive Dental Care, Medical University of Warsaw, 02-097 Warsaw, Poland

**Keywords:** periodontitis, IL-1, gene polymorphisms

## Abstract

Periodontitis is a chronic multifactorial inflammatory disease originating from microbial, environmental and genetic factors. The present study aimed to find an association of genetic polymorphisms at IL-1A^−889^ and IL-1B^+3953^ loci in Polish patients with stage III grade B periodontitis and periodontally healthy subjects. Fifty patients with stage III grade B periodontitis (tests) and thirty-five periodontally healthy controls were included in the study. To determine IL-1A and IL-1B gene polymorphisms, buccal swab-derived DNA obtained by means of the GenoType PST test was used. There were no statistically significant differences regarding the prevalence of IL-1A^−889^ or IL-1B^+3953^ alleles between groups. The frequencies of different IL-1A^−889^ genotypes did not differ significantly between groups. The IL-1B^+3953^ C/T genotype was significantly more predominant in periodontitis patients than in controls, whereas C/C genotype prevalence was significantly lower in the test group. Complex genotypes consisting of at least one IL-1A^−889^ and IL-1B^+3953^ T allele occurred significantly more frequently in subjects with periodontitis. Stage III grade B periodontitis may be associated with the IL-1B^+3953^ T allele and composite IL-1 polymorphism. Reduced susceptibility to periodontal disease was present in IL-1A^−889^ and IL-1B^+3953^ C/C homozygotic patients.

## 1. Introduction

Periodontitis is an inflammatory disorder with progressive attachment loss and bone destruction leading to possible tooth loss [1]. It is one of the most prevalent oral health problems, affecting 735 million adults worldwide [2]. Severe periodontitis affects 11.2% of the world’s population and is associated with male gender, age, tobacco smoking, a low level of socioeconomic status and education.

During recent years, there has been evidence of systemic effects of periodontal infection in relation to diabetes and atherosclerosis. Periodontal infection complicates glycemic control in diabetes and contributes to the risk of diabetes and associated complications. This infection results in an increase in levels of systemic pro-inflammatory mediators, which exacerbates insulin resistance. Diabetes promotes the occurrence, progression, and severity of periodontitis. It leads to a hyperinflammatory response to periodontal microbiota and also impairs the resolution of inflammation and repair, which leads to accelerated periodontal destruction. The cell surface receptor for advanced glycation end products and its ligands are expressed in the periodontium of individuals with diabetes and seem to mediate these processes. Treatment of periodontal infections should become an integral part of the management of diabetes, whereas glycemic control is a prerequisite for successful periodontal therapy. Periodontal infections are also considered an independent risk factor for atherosclerosis and its clinical sequelae, for example, cardiovascular diseases. These diseases have many pathogenic mechanisms in common. The severity and chronicity of periodontal disease provide a rich source of subgingival microbial and host response products and effects over a long time period. Such infections and chronic inflammations may influence the atherosclerotic process.

Periodontal diseases are characterized by the progressive destruction of soft and hard tissues of the periodontal complex, mediated by an interplay between dysbiotic microbial communities and aberrant immune responses within gingival and periodontal tissues. Putative periodontal pathogens are enriched as resident oral microbiota become dysbiotic and inflammatory responses evoke tissue destruction, thus inducing an unremitting positive feedback loop of proteolysis, inflammation, and enrichment for periodontal pathogens. Keystone microbial pathogens and sustained gingival inflammation are critical to periodontal disease progression.

Tissue destruction in the course of periodontitis results primarily from an immune-inflammatory response to periopathogens represented by species such as *Porphyromonas gingivalis*, *Treponema denticola*, *Tanerella forsythia* and *Aggregatibacter actinomycetemcomitans* [3]. The mechanisms of inflammation involve numerous pro-inflammatory mediators; hence, an increased risk of periodontitis may be related to the aberrant production of pro-inflammatory cytokines [4]. Interleukin-1 (IL-1) appears to be one of the most important molecules that is associated with the development and course of periodontal inflammation. IL-1 participates in a number of processes necessary to initiate and sustain an inflammatory response. It increases the production of adhesion molecules, facilitating leukocyte migration, stimulates the production of other inflammatory mediators and metalloproteinases, actives T and B lymphocytes, stimulates osteoblasts leading to bone resorption, and stimulates the programmed death of cells producing extracellular matrix, thus limiting the regenerative capabilities of tissues. IL-1 consists of 11 genes in the 430-kb fragment in the long-arm DNA of chromosome 2 in the 2q12-q21 region. These genes are responsible for the production of IL-1A and IL-1B. Generally speaking, IL-1 support increases the expression of adhesion receptors in the endothelial cells with the release of other cytokines, which leads to the amplification of neutrophil recruitment [5]. The salivary level of IL-1 receptor antagonist (IL-1Ra), an agent that joins to the IL-1 receptor, is also observed to be statistically higher in periodontitis patients than in controls [6].

It is a well-known fact that periodontitis is a polygenetic disease; hence, genetic polymorphisms associated with the increased production of pro-inflammatory mediators may potentially contribute to the greater severity of periodontitis. Variations in IL-1 genes were first associated with chronic periodontitis in Caucasians in 1997 [7]. A very recent study involving Bayesian methods to rate meta-analysis data on polymorphisms in IL-1 genes identified the IL-1A/rs1800587 and IL-1B/rs1143634 polymorphism as noteworthy biomarkers for chronic periodontitis susceptibility [8]. IL-1A and IL-1B genes displayed polymorphism at the IL-1A^−889^ and IL-1B^+3953^ loci [8,9,10]. Alleles associated with the C to T (allele 2) transition at these loci appear to be associated with an increased IL-1 gene expression [11,12]. The 1800587 polymorphism is recognized as a single nucleotide polymorphism (SNP). Even though the wild-type allele had an important minor allele frequency (MAF) in the global population of 0.7214, the mutant allele was behind increased IL-1A protein levels in gingival crevicular fluid in patients with periodontal disease [13]. The carriage of allele 2 was correlated with almost a four-fold increase in IL-1A protein levels. It was also reported that IL-1A 2/2 homozygotes (subjects double-positive for the mutant allele) had significantly elevated levels of IL-1B plasma [14]. Accordingly, a composite genotype, the so-called PST+ genotype, characterized by the simultaneous presence of IL-1A and IL-1B allele 2, was found to be related to greater severity of chronic periodontitis [7]. However, this association still remains a subject of controversy, and ethnicity is a dominant factor that impinges on cytokine gene polymorphism distribution. A current systematic review and meta-analysis showed that although IL-1B is a noteworthy cytokine biomarker in periodontitis development and progression, different IL-1A polymorphisms may have inverse actions in the pathogenesis of periodontitis.

The prevalence of genetic polymorphisms at the IL-1A^−889^ and IL-1B^+3953^ loci has not been studied in the Polish population so far. Therefore, our study was designed to determine the prevalence of genetic polymorphisms at the IL-1A^−889^ and IL-1B^+3953^ loci in adults with stage III grade B periodontitis and in periodontally healthy individuals. Typing of IL-1 gene polymorphisms may be of predictive value and might be useful for choosing the most effective therapy.

## 2. Materials and Methods

The present study obtained a positive affirmation from the institutional review board (KB/58/2011) and was carried out at the Department of Conservative Dentistry of the Medical University of Warsaw. All clinical procedures were achieved in accordance with the Helsinki Declaration of 1975, as revised in Tokyo in 2013. All patients have been informed about the study’s objectives, as well as possible risks and profits of participating in the study.

### 2.1. Inclusion Criteria

The inclusion criteria were as follows: (1) systemically healthy participants, diagnosed with stage III grade B periodontitis (tests) or diagnosed with healthy periodontium (controls) [15]; (2) presence of ≥20 teeth; (3) no professional dental debridement within 6 months before the examination; and (4) consent to participate in the study and signing the informed agreement form.

### 2.2. Exlusion Criteria

The exclusion criteria were as follows: (1) the presence of relevant systematic diseases that would affect their periodontal status (e.g., diabetes, blood disorders, or immunodeficiency); (2) being under systemic medications that can change the process of periodontal disease (e.g., antibiotics, steroids, anti-inflammatory drugs, immunosuppressants); (3) pregnancy or lactation and (4) smoking or the use of other tobacco products.

### 2.3. Study Design

The study consisted of a clinical and a laboratory component. A total of 10 non-study patients were enrolled for the calibration exercise. All measurements were performed by one therapist, thereby allowing intra-experimental comparisons of the values. The examiner underwent calibration training at the beginning of the study. Full-mouth PPD and CAL with an interval of 24 h between recordings were registered. A complete intraoral examination was conducted for all recruited participants using a graded periodontal probe (UNC probe 15 mm, Hu-Friedy’s, United States): The principal investigator evaluated: (1) the number of teeth present in the oral cavity; (2) the full-mouth plaque index (FMPI) according to O’Leary et al. [16] on four tooth surfaces as the number of surfaces with plaque divided by the number of all examined surfaces; (3) full mouth bleeding on probing index (FMBOP) according to Ainamo and Bay [17]: using the same probing pressure, sulcus bleeding was determined 30 s after probing and was assessed as either presence or absence at six sites per tooth (i.e., distobuccal, buccal, mesiobuccal, distolingual, lingual, mesiolingual); (4) probing pocket depth (PPD): pocket depth was measured at six sites at the tooth (mesio-labial, mid-labial, distal-labial, mesio-palatal/lingual, mid-palatal/lingual and distal-palatal/lingual sites) as the distance from the gingival margin to the bottom of the sulcus; (5) clinical attachment level (CAL) at six points of each tooth as the distance from the cemento-enamel junction (CEJ): this was recorded at 6 sites in a manner similar to PPD in relation to the cementoenamel junction. In connection with the clinical examination and evaluation of clinical diagnosis, panoramic and periapical X-rays were made [15]. Stage III periodontitis was identified when: (1) interdental CAL ≥ 5 mm; (2) radiographic bone loss extended to the mid-third of the tooth root and beyond; (3) tooth loss because of periodontitis was ≤4 teeth; (4) PPD ≥ 6 mm. Grade B periodontitis was evaluated based on dental radiograms as a percentage of root length divided by the age of the subject (% bone loss/age) when this value varied from 0.25 to 1.0. Healthy periodontium was defined as <10% bleeding sites with PPD ≤ 3 mm.

### 2.4. Polymorphism Analysis

IL-1A^−889^ and IL-1B^+3953^ polymorphisms were analyzed on the basis of a commercially available GenoType PST test (Hain Diagnostica, Germany; Greenstein and Hart 2002) according to a detailed description provided by the manufacturer. In brief, patients’ DNA was isolated from buccal swabs, and fragments of IL-1A and IL-1B genes were amplified by polymerase chain reaction (PCR). The obtained DNA fragments were then subjected to reverse hybridization with probes identifying alleles at the IL-1A^−889^ and IL-1B^+3953^ loci.

### 2.5. Statistical Analysis

The statistical analysis was performed using Statistica v. 13 (TIBCO Software Inc., Palo Alto, Santa Clara, CA, USA). Mean and standard deviations were calculated for each parameter. A test of normality was conducted using the Shapiro-Wilk test. The numeric demographic parameters that showed the normal distribution were analyzed by a *t*-test. The Mann–Whitney test was employed to identify the differences in clinical parameters between cases and controls. Differences between the test group and healthy controls in the frequencies of IL-1A^−889^ and IL-1B^+3953^ genotypes and alleles were assessed through Fisher’s exact test. For the determination of the strength of associations, odds ratios (OR) and their 95% confidence intervals (95% CI) were also evaluated. The results were considered statistically significant at least at *p* ≤ 0.05.

## 3. Results

A total of 50 patients with stage III grade B periodontitis stage and 35 periodontally healthy subjects were recruited. The test group included 26 females and 24 males with a mean age of 36.31 ± 8.27 years, and the control group included 19 females and 16 males with a mean age of 35.53 ± 6.88 years. Demographic and clinical characteristics are presented in Table 1.

Frequencies of the IL-1A^−889^ and IL-1B^+3953^ alleles in tests and controls are shown in Figure 1. There were no significant differences in the distribution of either IL-1 allele between both groups.

However, an analysis of frequencies of individuals carrying either allele revealed that the carriage rate of the IL-1B^+3953^ T allele in periodontitis patients was significantly increased (*p* = 0.049) (Figure 2).

An analysis of IL-1A^−889^ and IL-1B^+3953^ genotype frequencies revealed that the IL-1B^+3953^ C/T genotype was significantly more frequent in the group with periodontitis (*p* = 0.016), whereas the prevalence of the IL-1B^+3953^ C/C genotype was significantly lower (*p* = 0.029) (Figure 3). There was no difference in the distribution of the IL-1B^+3953^ T/T genotype. A similar trend was observed in the case of IL-1A^−889^ genotypes; the differences between groups, however, were at the borderline of significance (Figure 3).

An analysis of the composite PST+ genotype has shown that its frequency in tests and in controls was 26/50 (52.5%) and 10/35 (30.0%), respectively. This difference was statistically significant (*p* = 0.035). Furthermore, the PST+ genotype was associated with the prevalence of stage III grade B periodontitis (OR = 2.57; 95% CI, 0.97–6.99). The respective prevalence of IL-1A^−889^ and IL-1B^+3953^ double C/C homozygotes in periodontitis patients and in the healthy control group was 20/50 (40%) and 21/35 (60%). This difference was also statistically significant (*p* = 0.006) and suggests that this genotype may be associated with a decreased susceptibility to the disease (OR = 0.44; 95% CI, 0.16–1.07).

## 4. Discussion

The results of the present study confirm the role of IL-1 polymorphism in the pathogenesis of stage III grade B periodontitis. Since the new clinical classification of periodontitis has replaced the previous classification in chronic (CP) and aggressive periodontitis (AgP) quite recently, not many studies have implemented periodontal staging and grading in reporting on polymorphisms in interleukin genes.

The 1999 International Workshop on the Classification of Periodontal Diseases distinguished between aggressive periodontitis and chronic periodontitis. Consequently, chronic periodontitis affected a high proportion of adults and was characterized by slow progression. Aggressive periodontitis, on the other hand, was defined as a rare inflammatory condition characterized by rapid and severe destruction of connective tissue attachment and bone with minimal presence of the microbial deposits that affected younger individuals with familial aggregation. However, the 2017 World Workshop on Classification of Periodontal and Peri-Implant Diseases grouped aggressive periodontitis and chronic periodontitis under a single category, for which a new classification framework was updated to stages (I–IV) and grades (A–C). Staging was based on the severity and the extent of disease, while grading depended on the rate of progression, which was linked with risk factors. In spite of the recent modification in classification, we will refer to them as chronic periodontitis or aggressive periodontitis, as the majority of genetic studies were carried out under the old nomenclature.

The prevalence of IL-1A^−889^ and IL-1B^+3953^ genotypes in patients in this study was similar to that reported in earlier studies [18,19,20]. The prevalence of the composite PST+ genotype, which may be associated with periodontitis stage III grade B, was also comparable to that observed in populations covered by other research [21,22,23,24,25,26]. We did not find any association between periodontitis and either a single IL-1A^−889^ T allele or a single IL-1B^+3953^ T allele genotype. However, in line with the observations of Kornman et al. [7] and McDevitt et al. [27], an association with the composite PST+ genotype was noted. Furthermore, the prevalence of stage III grade B periodontitis was associated with the IL-1B^+3953^ C/T genotype. Thus, our findings are consistent with the observations of Zeng et al. [28], who reported a higher prevalence and carriage rate of the T allele in patients with chronic periodontitis, and Zuccarello et al. [29], who showed a higher prevalence of the IL-1B^+3953^ C/T genotype, together with a lower prevalence of the C/C genotype in periodontitis. On the other hand, Armingohar et al. [30] reported a higher prevalence of the 2/2 genotype combined with a lower prevalence of the 1/2 genotype. A current meta-analysis showed that IL-1B^−3954^ C > T polymorphism had a very strong relationship in the periodontitis risk with OR (95% CI), 0.66 (0.52–0.80) [10]. Quite interestingly, the framework genotype of IL-1A^−889^/IL-1B^+3953^ and IL-1B^+3954^ were significantly associated with the risk of dental peri-implant disease [31]. For IL-1A^−889^ polymorphism, the T allele, compared to the C allele, generated a strong expression of IL-1α, quite similarly to the TT genotypes compared to the CC genotype [13,32]. The T allele and CT genotype of IL-1B^+3954^ polymorphism were also correlated with a raised risk of dental peri-implant disease. We did not observe any significant association between IL-1B^+3953^ T/T genotype and periodontitis; this, however, may be due to a relatively low occurrence of this genotype in the evaluated sample. Interestingly, no association between the IL-1B^+3953^ allele 2-carrying genotype was seen in patients of Greek origin [25]. In some populations, the carriage rate of the IL-1B^+3953^ allele 2 was too low to draw conclusions about an association between IL-1B polymorphism and periodontitis [33,34]. In the present study, a tendency for the occurrence of stage III grade C periodontitis was revealed to be associated with the IL-1A^−889^ C/T genotype; this association, however, was at the borderline of significance. Other investigators found no association between the carriage rate of IL-1A^−889^ allele 2 or genotype and periodontitis [7,35].

The presented outcomes confirm the weight of the IL-1B^+3953^ T allele in the etiopathology of stage III grade B periodontitis. However, the role of the IL-1B^+3953^ C allele appears to be unclear. The frequency of the IL-1B^+3953^ C/C genotype was significantly decreased in the group with periodontitis. Furthermore, the dominance of double IL-1A^−889^ C/C and IL-1B^+3953^ C/C homozygotes in the group of patients with periodontitis was also significantly lower than in healthy controls. This strongly implies that the IL-1B^+3953^ C allele, and possibly the IL-1A^−889^ C allele, may conform to a resistance phenotype. A very recent meta-analysis showed that the IL-1A^−889^ C/T T allele had a beneficial association in the prophylaxis of periodontitis risk with an OR (95% CI) of 1.12 (0.99–1.25) [10]. In contrast, the IL-1A^−889^ C/T C allele also had a significant relationship with periodontitis progress with an OR (95% CI) of 0.75 (0.66–0.85).

It was reported that patients carrying the composite PST+ genotype have elevated levels of IL-1B in gingival crevicular fluid [36,37]. Following treatment, there was a reduction in IL-1B concentration in non-carriers of the composite genotype but not in carriers. Furthermore, the IL-1B^+3953^ T allele was shown to be associated with the increased secretion of IL-1B by monocytes following LPS stimulation [35]. Likewise, a strong trend towards an increased secretion of IL-1B by peripheral blood polymorphonuclear cells was described for carriers of allele 2 as compared to non-carriers [38,39]. It is, therefore, possible that the association between stage III grade B periodontitis and both IL-1B^+3953^ allele 2, as well as the IL-1 composite PST+ genotype, may be due to the increased formation of a pro-inflammatory cytokine, namely IL-1B. Conversely, double C/C homozygosity may possibly render patients less susceptible to severe forms of periodontitis because of the relatively lower production of pro-inflammatory cytokines. In a very recent study, heterozygous genotypes and SNP on the IL-1A^−889^ and IL-1B^+3954^ gene loci did not influence CAL after non-surgical therapy and 2 years of maintenance [40]. Wild-type patients showed greater improvements in CAL. Taking everything into account, IL-1B may be an important therapeutic target for periodontitis. Polymorphism (rs2234663) of IL-1Ra was found to contribute to major periodontitis receptivity [41]. IL-1Ra can stop the biological action of IL-1B without triggering any intracellular signaling. IL-1rA-loaded dextran/PLGA microspheres efficiently inhibited gene expression of pro-inflammatory factors induced by IL-1B in human gingival fibroblasts [42]. Hence, the microspheres may be a very good aspirant for the treatment of periodontal disease.

Variations in the results of the present study and the above-cited studies may have been due to different criteria in the selection of patient and control groups. In the present study, we studied non-smoking, healthy, adult Caucasians using the current classification of periodontal diseases. It was previously shown that ethnicity may greatly influence cytokine gene polymorphism distribution [43]. In a Caucasian population from England, the T allele had an MAF of 0.3132, whereas in an Asiatic population from Tokyo, the MAF value was 0.0625. Kobayashi et al. [44] assessed 100 Japanese patients with periodontal disease and 100 healthy controls. The authors found a percentage of 3% for the T allele in both groups. No T/T homozygous genotype was observed. IL-1 gene polymorphisms did not constitute a common risk factor for periodontitis (*p* > 0.05). However, it was suggested that the distributions of IL-1B^+3954^ genotypes and IL-1A^+4845^ and IL-1B^+3954^ haplotypes were unique to the Japanese patients with rheumatoid arthritis and periodontitis. In a mixed population, a higher T allele frequency was observed as compared with the Asian population [45]. However, the IL-1B (rs1143634) SNP could not be found as a risk predictor for chronic periodontitis. The authors concluded that synergistic interaction of the CT/CC genotypes of NLRP3 (rs4612666) SNP with ageing and smoking habits might potentially undermine the pathogenic tract of periodontal disease. In the present study, only non-smokers were evaluated, which may have had an impact on the outcomes. Accordingly, an association was previously reported between the severity of periodontitis and the composite genotype in non-smokers [7,46] and in former light to moderate smokers [27,47]. In another study, the extent of CAL loss was significantly associated with the composite genotype of IL-1A/-1B in smokers, which ensured the information that the polymorphism of IL-1 shows an interaction with smoking, the main modifiable risk factor of periodontitis [48]. Regardless of the status of the IL-1 genotype, smoking was associated with a higher prevalence of periodontitis.

Some limitations of this study should be considered. First, there was a limited number of included subjects, especially controls with healthy periodontium. It is widely known that sample size may influence polymorphism. In addition, periodontitis is a multifactorial disease, and many local and environmental factors are involved in its progression, apart from possible genetic predispositions. In the present study, groups were matched in terms of age and sex, and only systematically healthy and non-smoking subjects were involved. However, they differed significantly in terms of oral hygiene status. Finally, microbiological and immunological analyses were not carried out. Given this, further bigger and well-designed studies among people of different ethnicities and with various periodontal diagnoses are needed to elucidate the present findings. 

## 5. Conclusions

Within the limits of the present study, it can be concluded that the occurrence of stage III grade periodontitis might be associated with IL-1B^+3953^ T allele and composite IL-1 polymorphism, and a decreased susceptibility appeared to involve IL-1A^−889^ and IL-1B^+3953^ C/C homozygosity in adult Poles. More research studies are needed to better understand the weight of gene polymorphisms in periodontitis.

## Figures and Tables

**Figure 1 ijerph-19-14687-f001:**
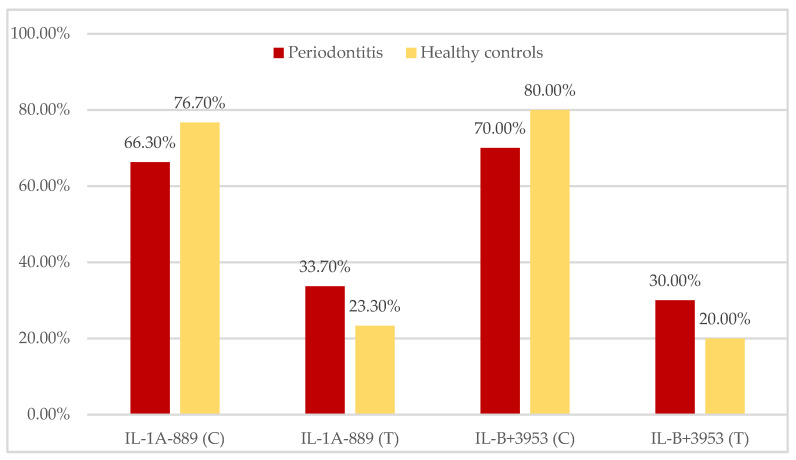
IL-1A^−889^ and IL-1B^+3953^ allele frequencies in periodontitis patients and healthy controls.

**Figure 2 ijerph-19-14687-f002:**
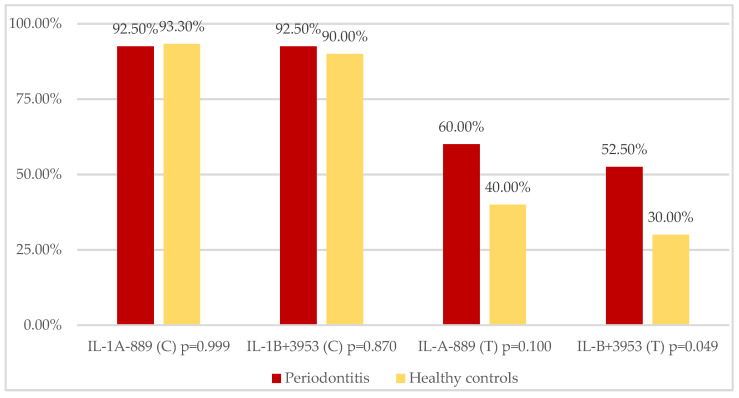
Frequencies of individuals carrying IL-1A^−889^ and IL-1B^+3953^ C or T allele.

**Figure 3 ijerph-19-14687-f003:**
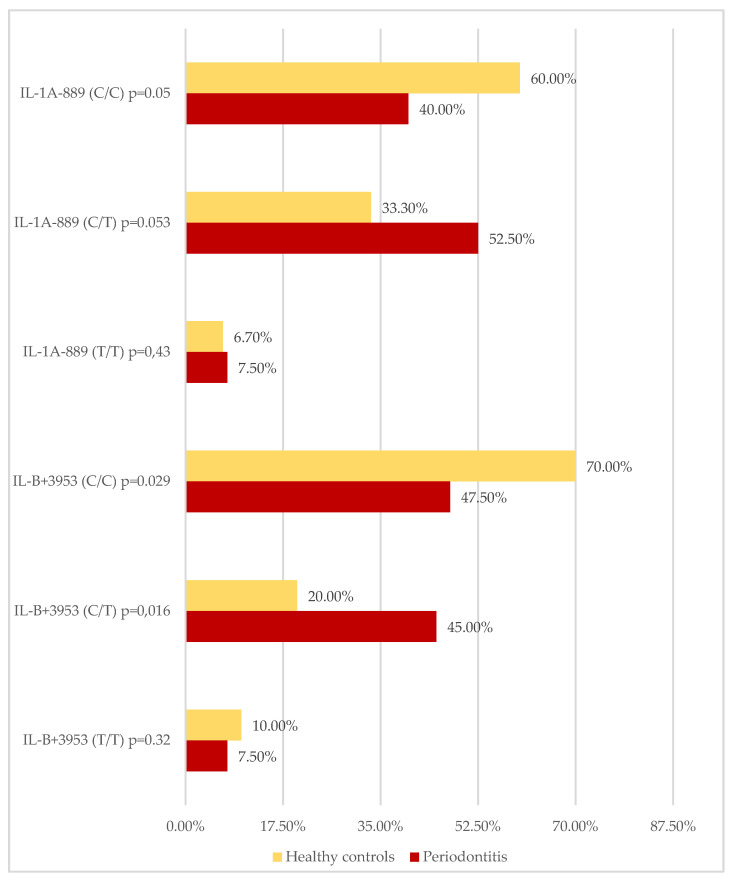
Frequencies of IL-1A^−889^ and IL-1B^+3953^ genotypes in periodontitis patients and healthy controls.

**Table 1 ijerph-19-14687-t001:** Demographic and clinical characteristics of tests and controls.

Variable	Tests	Controls	*p*
Age mean ± SD (years)	36.31 ± 8.27	35.53 ± 6.88	0.431
Sex F/M n (%)	26/24 (52/48)	19/16 (54.2/45.8)	0.425
Number of teeth	22.66 ± 1.53	23.12 ± 1.10	0.406
PPD (mm)	4.32 ± 1.68	1.45 ± 0.66	<0.001
CAL (mm)	3.68 ± 1.87	0.083 ± 0.22	<0.001
FMPI (%)	20.51 ± 6.57	6.59 ± 5.69	<0.001
FMBOP (%)	38.82 ± 19.02	15.25 ± 11.65	<0.001
IL-1A CC/CT/TT n (%)	16/40 (40.0%) 21/40 (52.5%) 3/40 (7.5%)	18/30 (60.0%) 10/30 (33.3%) 2/30 (6.7%)	0.051 0.053 0.431
IL-1B CC/CT/TT n (%)	19/40 (47.5%) 18/40 (45.0%) 3/40 (7.5%)	21/30 (70.0%) 6/30 (20.0%) 3/30 (10.0%)	0.029 0.016 0.322

Abbreviations: PPD—probing pocket depth; CAL—clinical attachment loss; FMPI—full mouth plaque index; FMBOP—full mouth bleeding on probing index.

## Data Availability

Research data is available on request through the authors themselves. Aniela Brodzikowska should be contacted to request the data (aniela.brodzikowska@wum.edu.pl).

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
