# Peer review of "Association between IL-1 Gene Polymorphisms and Stage III Grade B Periodontitis in Polish Population"

_ijerph, 2022, doi:10.3390/ijerph192214687_

Round 1

Reviewer 1 Report

1# In the title the author incorporate ‘Title’ by mistake, I imagine!! Need to polish it.

2# Need to reduce word count in Abstract. The author is suggested to check author guideline for this section. Furthermore, they used headings in the abstract while it is instructed not to do. The rule is as mentioned ‘The abstract should be a single paragraph and should follow the style of structured abstracts, but without headings’. The author needs to be more careful when writing a scientific manuscript. This type of error is wastage of time for the respective reviewer.

3# Again, the first line of the abstract states the aim of this work. The author should present the background of this research then aim. Need to be rewritten.

4# The author should follow one format among IL-1A and IL-1B Or, IL-1α and IL-1β throughout the manuscript.

5# The correlation between IL-1 and periodontitis is well established. However, the author aimed to check it among Polish people. In that aspect this manuscript has merit, but it will lose global concern, as well. Anyway, the author is suggested to add ‘in Polish people’ to the Title ans Abstract.

6# In this experiment, only 50 samples are examined. The sample size is very low. Do they have any reference of acceptable sample size?

7# Stage III grade B periodontitis is equivalent to acute or chronic periodontitis? The author need to mention it in their manuscript.

8# The author mentioned that the carriage rate of IL-1B+3953 T allele in periodontitis patients was significantly increased in Figure 2. However, there is no sign of significance in the figure, as well as in the figure legend. Does the result of figure 3 is significant? If so, then needs to add significant sign also.

9# They need to compare their result obtained from Polish people with other existing data sets.

10# There are many typos in this manuscript. Careful editing is required.

Reviewer 2 Report

In the manuscript entitled “Association between IL-1 gene polymorphisms and stage 2 III grade B periodontitis” the authors aimed to find out an eventual association of genetic polymorphisms at IL-1A-889 and IL-1B+3953 loci in patients with stage III grade B periodontitis and periodontally healthy subjects. The study is interesting and the manuscript well presented. Here some suggestions to improve the quality of the manuscript:

INTRODUCTION

_ I suggest explaining in this paragraph the concept of dysbiosis and the disruption of the balance between oral bacteria and the immune system.

_ It would also be appropriate to explain how periodontal disease also involves systemic inflammation and is related to other systemic diseases (in particular cardiovascular and diabetes).

_ The function of IL-1 and its sub-variants in inflammation in general should be better explained.

MATERIALS AND METHODS

_ I simply suggest dividing it into subparagraphs (e.g. study design, inclusion / exclusion criteria, polymorphism analysis, statistical analysis, etc.).

_ Why were 50 patients with periodontal disease and only 35 healthy controls selected?

Round 2

Reviewer 1 Report

In Figure 3, correct the p values. It would be 0.43 not 0,43. Check another values also.